# Perceived Social Support and Engagement in First-Year Students: The Mediating Role of Belonging during COVID-19

Jorge Maluenda-Albornoz [1,*], José Berríos-Riquelme [2] , Valeria Infante-Villagrán [3,4] and Karla Lobos-Peña [5]

1  Facultad de Psicología y Humanidades, Universidad San Sebastián, Sede Concepción, Concepción 4080871, Chile
2  Departamento de Ciencias Sociales, Universidad de Tarapacá, Iquique 1113749, Chile
3  Departamento de Psicología, Facultad de Ciencias Sociales, Universidad de Concepción, Concepción 4070386, Chile
4  Departamento de Psicología, Facultad de Ciencias de la Salud, Universidad Católica de Temuco, Temuco 4813302, Chile
5  Escuela de Psicología, Facultad de Educación y Ciencias Sociales, Universidad Andrés Bello, Concepción 4300004, Chile
*  Correspondence: jorge.maluenda@uss.cl

**Abstract:** Academic engagement and the conditions that favor it have become relevant in recent decades due to their relationship with academic performance, well-being, and university permanence. Variables such as perceived social support and sense of belonging are relevant aspects of social integration to promote engagement. Evidence shows both variables predicting engagement. In addition, the available evidence suggests this possible mediating role, which requires further analysis in freshmen in the emergency teaching context due to the COVID-19. The present study aims to evaluate the mediating role of sense of belonging in the relationship between perceived social support and engagement in university students. Results showed significant predictive relationships between social integration variables and engagement and showed the mediating role of sense of belonging in the relation between perceived social support and the three types of engagement. These results suggest the relevance for degree programs to consider these social integration variables as a key element for university freshmen.

**Keywords:** engagement; sense of belonging; social support; freshmen; university

## 1. Introduction

Starting a university degree is a difficult experience for students. They face finding themselves, constructing their identity, and trying to overcome the transition from secondary education to university education with the complications that this entails [1]. The student must learn the formal and informal codes and regulations of the institution in which they are located; they have to establish new interpersonal relationships—with peers, professors, and other officials—that allow them to have a social network within the university.

The construction of support networks was abruptly affected during the first year of the pandemic, with severe changes in different areas. This phenomenon significant impacted students, especially freshmen entering higher education [2]. Students were working with peers and teachers through electronic devices, trying to complete academic tasks and still enjoy a social university experience. These changes generated differences in the habitual norms of the initial university experience, since it was necessary to rethink teaching innovation strategies, change the programmed methodologies, and allocate more institutional resources; above all, there was a radical change in the methods of motivating and supporting students.

The basic psychological need for relationships was affected by the confinement and isolation in this new academic space of interaction. Students were trying to resolve problems, communicate, and negotiate while working on teams with communication delays and lack of real physical contact. Recent research made during the pandemic analyzed social integration variables, finding that social interaction, participation and feedback, and close relationships with peers and teachers, were the most valued teaching–learning practices and had the greatest impact in this emergency virtual context [3]. At the same time, several studies have shown that confinement, isolation, and lack of social interaction decreased the motivation and participation of students [4,5].

In this scenario, where social integration variables have been compromised, student's engagement could be affected, given that engagement is built on the conception of the interactions that the student has with the educational context [6]. In this regard, it is important to consider engagement in first-year university students, since it has been shown to be important in academic variables such as retention and performance [7].

Few studies have analyzed the variables of social integration and their relationship with engagement in first-year students during confinement. This issue deserves attention, given that the confinement would have affected the variables of social integration and, therefore, would have impacted the development of engagement in this population.

## 1.1. Engagement in University Context

In the university context, engagement is a high motivational state that manifests itself in student behavior [3]. When students are in this motivational state, they show interest in educational activities, and they make an effort and dedicate time to learning [8–11]. By understanding engagement as a highly motivational state, it is possible to operationalize students' attitudes in three dimensions: behavioral, affective, and cognitive [10].

The behavioral dimension includes all the behaviors that are observed in students that are interested in learning [11], such as participation, collaboration, performance, and interest [12]. The cognitive dimension refers to the purpose of learning and is related to the thoughts, beliefs, and perceptions that students have regarding the effort that academic work requires. Therefore, it includes characteristics such as motivation, critical thinking, and understanding of complex ideas, among others [12]. Finally, the emotional dimension is related to the attitude and feeling of students towards the institution they belong to. Some criteria that are evaluated in this dimension are the perception of relations with peers, teachers, and the institution itself [12].

In the context of the pandemic and according to Junt and Lee (2018), in online education, behavioral engagement is related to asking questions and communicating through electronic devices; cognitive engagement is understood as the effort that learners make to develop specific skills or understand online subjects; and emotional engagement is related to the positive feeling towards teachers, students, and the subjects they take.

When students are "engaged", they participate and collaborate in academic, social, and extracurricular tasks linked to their learning. Therefore, they achieve positive academic results. Furthermore, when students engaged, they are perceived to have more energy and willingness to make the effort that is required to perform complex tasks and to develop complex skills [13,14]. Additionally, they tend to manifest effort, persistence, and prosocial behavior in class (behavioral engagement). Consequently, when students engage, they show positive characteristics such as high interest in the topics they are learning, enthusiasm, concentration, strategic learning, and self-regulation (cognitive engagement). On the contrary, when students are engaged, they show a decrease in the levels of anxiety and boredom (emotional engagement) [13,15,16].

Research on Study Engagement has shown relevance through its relationships with a several variables linked to the teaching–learning process in university students [17]. Study engagement has been related with burnout, present and future academic performance [18], wellbeing [19,20], academic satisfaction [20,21], dropout intention [22,23], and

dropout [24,25]. For these reasons, it is imperative to analyze the social integration variables that are related to engagement in first-year college students.

*1.2. Social Integration Factors Related to Engagement*

In the university context, social integration is related to the adaptation that students achieve in the institution through daily experiences [26]. Social integration variables such as perceived social support and sense of belonging have been deeply studied in the past few years [27,28].

On the one hand, perceived social support is defined as the evaluation of students regarding the resources the institution makes available for social support. This considers both the quantity and quality of the social support [29]. Therefore, perceived social support is a subjective and personal evaluation [30]. In this sense, the subjective element of the evaluation is key, since it will depend directly on the personal needs that students have concerning social support or social contact [31].

It has been demonstrated that in an educational environment, perceived social support is related to the level of engagement that students have [23]. It has been observed that perceived social support and the interpersonal relationships that students establish with their peers, teachers, and members of the university campus are fundamental to develop a sense of belonging [32]. Thus, the sense of belonging is related to the process of integration of a person with the organization they belong to. Consequently, a student that has a feeling of belonging towards the institution will also have a high perception of social support.

Sense of belonging refers to the perception that students have of being part of the educational organization [33]. Having a high perception or sense of belonging means that students feel valuable and respected in their own educational program [34–38]. Furthermore, perceiving a strong or special bond between students and others makes them feel that they are part of a community, even when going through difficult moments or when facing challenges [37]. Therefore, the sense of belonging is useful for students, since it means that they feel accepted within the educational system, especially by peers and teachers [39].

To develop a high level of sense of belonging requires to have positive interactions in a stable context, such that students can feel part of a community within the university [39]. Therefore, students who feel comfortable with the formal academic learning environment and the social and cultural environment inside the institution have a sense of belonging that translates into a desire to commit to their studies and to achieve their academic goals [40]. Since the sense of belonging is such an important variable for students' education, its influence on the level of engagement has been deeply studied during the pandemic, as there has been a lack of social interaction that could have a negative effect on belonging [3]. Moreover, during the pandemic, it has been shown that sense of belonging is a predictor of cognitive and emotional engagement [3].

Some authors have argued that in higher education institutions, sense of belonging is a fundamental process that emerges from a collective one and that promotes the development of engagement [38]. This is because belonging involves the development of a student's sense of identification with their university. Therefore, it can be inferred that belonging predicts the level of engagement in the academic context [41]. Belonging can be determined by the similarity and connection that students perceive with respect to their immediate academic community [42]. It has also been shown that sense of belonging is a predictor of behaviors such as respecting the rules or assuming more functions than the mandatory ones [43].

Although there is evidence of the relationship between perceived social support, sense of belonging, and engagement, there is still little evidence evaluating a mediating model of this relationship. It has been found that the support of peers and parents influences the sense of belonging that students have towards their institution and is linked to a greater institutional engagement [44]. This leads to propose a scenario where perceived social support influences engagement, and sense of belonging contributes to explain this relation by a mediating effect. This is consistent with studies that have evaluated the mediating



role that sense of belonging has in the relationship between resilience and engagement in higher education students [45].

The present study aims to evaluate the mediating role of sense of belonging in the relationship between perceived social support and engagement in university students. The diverse evidence available suggests this possible mediating role, suggesting the need for analysis of first-year students during the mandatory confinement period.

## 2. Materials and Methods

### 2.1. Design

A cross-sectional associative-predictive design was used by testing mediation models [46]. Dependent variables were three dimensions of engagement (cognitive, emotional, and behavioral). Independent variables were perceived social support and perceived social isolation. The mediator of these relationships was the sense of belonging.

### 2.2. Participants

A convenience sampling method was used because of the restrictions imposed by the COVID-19 pandemic. Participants were 700 freshmen enrolled in 2020 in a Chilean university. The distribution of the sample was 280 men (40%), 418 women (59.71%), and 2 students who identified with another preference (0.29%). Student's age average was 18.4 years (SD = 1.7; minimum = 17; maximum = 32).

### 2.3. Instruments

The electronic questionnaire consisted of 27 items taken from 4 culturally and linguistically adapted instruments for the Chilean context. The response was made using a Likert-type scale of 1 to 7 points (1 = maximum disagreement; 7 = maximum agreement). The subsections of the questionnaire were as follows.

The University Student Engagement Scale consists of 15 items and was created by Maroco et al. [25]. This instrument has an adapted version for Chilean university students [24]. It measures engagement by 3 dimensions: Interest (5 items), Effort (5 items), and Participation (5 items). Adaptation and validation results in the university Chilean context showed good results for a bifactorial structure, with one general factor and three subfactors (RMSEA = 0.047 [95% CI: 0.040–0.055]; $\chi^2$ = 210.276, $p$ < 0.001; CFI = 0.967; TLI = 0.954) as well criterion validity and reliability ($\omega$ = 0.843; $\alpha$ = 0.841).

Membership factor was extracted from the Organizational Identification Questionnaire with Study Centers created by Yáñez et al. [47] and adapted to Chilean university students [39]. In this research the Belonging dimension was used and was made up of 4 items. The psychometric study in the adapted version showed good results (RMSEA = 0.028 [95% CI: 0.000–0.085]; $\chi^2$ = 3.126, $p$ = 0.20; CFI = 0.999; TLI = 0.999; RSMR = 0.005) as well as reliability ($\omega$ = 0.834; $\alpha$ = 0.815).

The perceived social support factor was inspired by the conceptual proposal of Biasi, Vicenzo and Patrizi [48]. This factor measures student's perception of having a supporting network in their educational community (degree), focusing on peers and professors, when it is needed. The factor was made up of 4 items, with good psychometric properties for a one factor structure (RMSEA = 0.072 [95% CI: 0.041–0.131]; $\chi^2$ = 11.616, $p$ = 0.003; CFI = 0.997; TLI = 0.992; RSMR = 0.011) as well as reliability ($\omega$ = 0.823; $\alpha$ = 0.798) [3].

### 2.4. Procedure

All new students enrolled in 2020 were invited to participate openly and voluntarily through an official email sent by the university's academic direction office and were equivalent to 16.78% of the total population.

For ethical safeguards, the Declaration of Helsinki was considered as a reference [49]. The students reviewed and signed an informed consent designed by the research team and approved by the Ethics Committee, which led them to the instrument in electronic format. The students who did not want to participate were excluded from the research.

Participation did not imply any compensation and was voluntary. The data collection was made in the first semester of 2020.

### 2.5. Analysis

Data analysis consisted of testing mediation models following the proposed procedures by Baron and Kenny [46]. Indirect, direct, and total effects were tested using the Maximum Likelihood estimator with 95% confidence interval. Correlation analysis was made using Pearson's r coefficient. Plots were made considering parameter estimates. All the analyses were made using Mplus version 7.1.

### 3. Results

Before performing the analyses, the distribution of the different study variables was tested (Table 1). None of the variables met the assumption of normality, which is why Spearman's Rho coefficient was used to analyze the correlations.

**Table 1.** Descriptive Statistics.

| z | Sense of Belonging | Cognitive Engagement | Behavioral Engagement | Affective Engagement | Social Support |
|---|---|---|---|---|---|
| Valid | 700 | 700 | 700 | 700 | 700 |
| Missing | 0 | 0 | 0 | 0 | 0 |
| Mean | 19.98 | 28.32 | 26.63 | 27.52 | 25.09 |
| Std. Deviation | 5.04 | 4.64 | 5.21 | 5.10 | 7.87 |
| Minimum | 4.00 | 5.00 | 5.00 | 8.00 | 6.00 |
| Maximum | 28.00 | 35.00 | 35.00 | 35.00 | 42.00 |

The results showed statistically significant correlations between all the variables studied (Table 2). The observed correlations fluctuated between medium to strong values, where the strongest association was found between affective engagement and sense of belonging. On the other hand, the weakest relationship observed was between cognitive engagement and sense of belonging.

**Table 2.** Correlations between variables.

| | | Social Support | Cognitive Engagement | Behavioral Engagement | Affective Engagement | Sense of Belonging |
|---|---|---|---|---|---|---|
| Social Support | Spearman's Rho | — | | | | |
| | *p*-value | — | | | | |
| Cognitive Engagement | Spearman's Rho | 0.305 *** | — | | | |
| | *p*-value | <0.001 | — | | | |
| Behavioral Engagement | Spearman's Rho | 0.344 *** | 0.48 *** | — | | |
| | *p*-value | <0.001 | <0.001 | — | | |
| Affective Engagement | Spearman's Rho | 0.404 | 0.376 *** | 0.417 *** | — | |
| | *p*-value | <0.001 | <0.001 | <0.001 | — | |
| Sense of belonging | Spearman's Rho | 0.532 *** | 0.317 *** | 0.432 *** | 0.692 *** | — |
| | *p*-value | <0.001 | <0.001 | <0.001 | <0.001 | — |

*** $p < 0.001$.

Following the main hypothesis of the study, the mediating role of sense of belonging in the relationship between perceived social support and engagement was tested. To test this hypothesis, three structural equation models were developed that evaluated the indirect effect of sense of belonging on the relationship between perceived social support and the different dimensions of academic engagement. In Table 3 total, direct, and indirect effect is reported.

**Table 3.** Standardized estimations of total, direct, and indirect effects.

| Effect | Cognitive Engagement Mediation Model | Affective Engagement Mediation Model | Behavioral Engagement Mediation Model |
|---|---|---|---|
| Indirect | 0.054 *** | 0.218 *** | 0.106 *** |
| [95% CI] | [0.028 to 0.079] | [0.184 to 0.251] | [0.076 to 0.136] |
| Direct | 0.14 *** | 0.052 ** | 0.119 *** |
| Total | 0.194 *** | 0.27 *** | 0.225 *** |

** $p < 0.01$, *** $p < 0.001$.

In the first model evaluated, results show that perceived social support predicts cognitive engagement in a statistically significant way ($\beta = 0.194$, $z = 9.291$, $p < 0.001$). When evaluating the mediating role of the feeling of belonging, it is evident that the relationship between both variables decreases in a statistically significant way, but the direct effect continues to be statistically significant ($\beta = 0.14$, $z = 5.783$, $p < 0.001$). In this way, it is possible to affirm that the feeling of belonging partially mediates (27.83%) the relationship between perceived social support and cognitive engagement ($\beta = 0.054$, $z = 4.061$, $p < 0.001$).

In the second model evaluated, the total effect of social support on affective engagement was statistically significant ($\beta = 0.27$, $z = 12.196$, $p < 0.001$). When assessing the role of sense of belonging, it is possible to assert that the indirect effect is statistically significant ($\beta = 0.052$, $z = 2.482$, $p < 0.01$) and, at the same time, the direct effect decreases but continues to be statistically significant ($\beta = 0.218$, $z = 12.683$, $p < 0.001$). Considering these values, it is possible to assert that we are in the presence of a partial mediation, where 19.25% of the effect of perceived social support on affective engagement can be explained through the feeling of belonging variable.

The third model evaluated analyzed the effect of perceived social support on behavioral engagement ($\beta = 0.225$, $z = 9.652$, $p < 0.001$). For its part, the indirect effect is statistically significant ($\beta = 0.106$, $z = 6.911$, $p < 0.001$), and the direct effect continues to be statistically significant ($\beta = 0.119$, $z = 4.518$, $p < 0.001$). These results show partial mediation, where 47.11% of the effect of perceived social support on behavioral engagement is channeled through the feeling of belonging.

## 4. Discussion

The pandemic forced higher education institutions to close due to mandatory confinement and start implementing online classes. The relationships and interaction at the university and inside the classroom changed, as well as the social and educational experience. Professors had to use new methodologies and institutions had to use new strategies to provide support to their students, especially those who entered the university in 2020.

The present study tested the mediating role of sense of belonging in the relationship between perceived social support and engagement in a sample of first-year university students during the start of the pandemic in order to contribute to understand relationships between these variables in the context described above.

The observed correlations allow us to indicate that there is a strong relationship between perceived social support and the sense of belonging, the first being a significant predictor of the second, in accordance with what was observed in previous research in a regular context and in virtual education during the pandemic [23,24,32]. This implies that,

by perceiving greater social support, a greater sense of belonging is also felt. In this way, maintaining better relationships with peers and teachers affects belonging experienced by students.

Perceived social support showed a moderate relationship with three types of engagement and was also a significant predictor, in accordance with previously reported research in a regular context and in virtual education through the pandemic [23,24,50]. The link between perceived social support regarding behavioral and cognitive engagement may be because having support within an academic activity can encourage students to be interested, involved, and persevere in the academic tasks they face. The strongest relationship, however, was obtained with respect to affective engagement, which may be due to the importance of positive relationships in the classroom to generate favorable and satisfactory work environments that stimulate student motivation.

Sense of belonging showed a moderate and predictive relationship with respect to cognitive and behavioral engagement and a strong relationship with respect to affective engagement, in accordance with what was reported in previous research in a regular context and in virtual education during the pandemic [23,24]. The relationship observed between the sense of belonging with respect to cognitive and behavioral engagement may be linked to people's behavior when part of a reference group [42,43]. Students who maintain stronger ties to their degree and better align with social and academic degree interests may experience greater interest, engagement, and participation. The greater relationship between the sense of belonging and affective engagement, in line with what was previously stated, may be because a student that experiences sense of belonging develops a positive relationship with peers and teachers and perceives a better work environment. As a result, they perceive a different academic workspace, which can affect their level of motivation.

Regarding the mediating role of the sense of belonging in the relationship between perceived social support and the different types of engagement, it was observed that the sense of belonging mediated the relationship with respect to the three types of engagement. In all cases, the mediation was partial and explained a good part of the relationship between the variables. These results are indicative that there is a direct influence of perceived social support on the different types of engagement, but that part of the relationship is explained by the presence of a sense of belonging. In other words, the effect that a greater perception of social support exerts on the engagement occurs because it affects it directly but also because it influences the students to feel a greater sense of belonging.

In this way, a student exhibits greater behavioral (participation, adherence, etc.), cognitive (more effort and intellectual involvement), and affective (enjoyment, positive attitude, etc.) engagement because they perceive more support from peers and teachers, but also because this support makes them feel a greater sense of belonging to their educational community.

The sense of belonging mediated the relationship between social support and affective engagement to a greater extent than the other types of engagement. According to what was previously stated, this effect may be linked to a greater preponderance of positive social relations and a favorable socio-academic climate to produce positive effects on the affections of the students around their studies and academic tasks.

The results described and analyzed above suggest that, in an emergency virtual education context as described in this article, perceived social support is a predictor of academic engagement. In the same way, the sense of belonging is configured as a measure that allows partially explaining this relationship, showing how supportive relationships can influence commitment, but also that they can influence belonging, and through this, the motivational states of the students.

## 5. Conclusions

The present study aimed to evaluate the mediating role of sense of belonging in the relationship between perceived social support and engagement in university students. Re-

sults showed the mediating role of sense of belonging and additionally showed differences in the percentage of mediation for each type of engagement.

It can be concluded that the sense of belonging acts as a partial mediator of the relationship between perceived social support and academic engagement. In addition, it is observed that mediation is greater in the case of the relationship between perceived social support and affective engagement, which highlights the importance of the effect of social relationships on the motivational experience of the university student.

The relationships observed and the mediating role were appreciated in a context of virtual education, in which the way of establishing interaction with peers and teachers was mediated by technological supports, the use of software with different social possibilities, and delays in communication, among other factors. The results presented occurred under these conditions, which allows reaffirming the relevance of these variables in influencing the motivation of students in that context. However, it is not possible to determine whether these variables acquire greater or lesser relevance with respect to a regular context because the study is limited by the lack of a previous valid basis for comparison. This allows an interesting line of research to emerge to elucidate this question.

Another limitation of this study has to do with the sampling that reduces the sample to a portion of university students. This restricts the possibilities for analyzing differences by type of degree and type of university and/or generalizing the results to the university population. This type of disaggregation of the results emerges as a future line of research, as well as differences by semester of study and gender, among others, which could be relevant.

These results have possible relevant applications to higher education institutions and can be useful to guide the support provided by higher education institutions to promote the academic engagement of students in virtual educational contexts. Generating connection strategies between students can favor perceived social support and thus engagement. However, strategies that allow students to connect with each other and with their teachers, in an integrated educational community that allows students to feel part of it, can be a deeper and more powerful strategy that encourages academic engagement. On the other hand, interventions to promote post-pandemic adaptation should consider the strengthening of these constructs among students, especially new entrants and those who took online classes in their first two years of university.

**Author Contributions:** Formal analysis, J.B.-R.; Funding acquisition, J.M.-A.; Investigation, J.M.-A., J.B.-R. and V.I.-V.; Project administration, J.M.-A.; Writing—original draft, J.M.-A.; Writing—review & editing, J.B.-R., V.I.-V. and K.L.-P. All authors have read and agreed to the published version of the manuscript.

**Funding:** This publication received funding from Universidad San Sebastian through project VRID_DocI22/01u.

**Institutional Review Board Statement:** The study was conducted in accordance with the Declaration of Helsinki and approved by Ethics Committee of the Research and Development Vice-presidency of Universidad de Concepción for studies involving humans (Certificate: CEBB 645-2020, April 2020).

**Informed Consent Statement:** Informed consent was obtained from all subjects involved in the study.

**Data Availability Statement:** Data supporting the reported results can be found at https://drive.google.com/file/d/127bIfoc3n6HK8fyMDJKQL-YzgCQzsp1W/view (accessed on 16 November 2022).

**Acknowledgments:** The authors are grateful to the Universidad San Sebastián for providing the necessary resources to publish this research through project VRID_DocI22/01u.

**Conflicts of Interest:** The authors declare no conflict of interest.

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
