# Peer review of "Perceived Social Support and Engagement in First-Year Students: The Mediating Role of Belonging during COVID-19"

_sustainability, doi:10.3390/su15010597_

Round 1

Reviewer 1 Report

Dear Authors,

First of all this manuscript about "Perceived social support and engagement in first-year students: the mediating role of the feeling..." and this is important especially due to Covid-19 era.

Well writen article. You used quantitative research method. Please explain quantitative research method first. Need to cite methodology articles and books in this part when you explain methods. You need to improve methodology part. This is important part of the research and need to be explain clearly.

Please reorganize the conclusion part too. Step by step. Give your results, explain problems and give suggestions. 

After the minor revisions, I am glad to say the manuscript will be ready for publication. 

Congrats!

Author Response

Dear reviewer After the exhaustive review of our article based on the comments provided, we inform you: 1.- We have revised the discussion and conclusions section, improving the way of writing and organizeation. 2.- We have added appropriate references for the use of the methods used according to your request. Thank you very much for the comments that were very useful to improve the manuscript.

Reviewer 2 Report

Dear Authors,

I have read the manuscript with much interest. It deepens some relevant constructs for understanding academic engagement and, in turn, academic performance.

The manuscript is adequately written, the literature review is complete and concise, and the data analyses are consistent with the research hypotheses.

I suggest some revisions to improve the quality of the work before publication:

1. If you have tested SEM, you have to report the fit indexes for each of them, besides the betas. Moreover, the mediational analysis should be presented through a table reporting standardized betas, C.I., and p for indirect and direct effects. A figure representing the model(s) could help the reader to visualize the relationships. 

2. I have detected some typos in the text, and some Spanish expressions: please revise accurately the text, a check by an English speaker could be useful.  

3.Please avoid causal language (i.e., predictive), as the cross-sectional study does not allow to infere causal relationships

Author Response

Dear reviewer, We have taken your comments into consideration and have fully incorporated them: 1- Indeed we have used SEM for the analysis. On this occasion we have included the fit indices, standardized betas, C.I and p, for each model. 2- We have carried out an in-depth review of the language used and corrected errors in writing and use of the language, as well as expressions of causality. We thank you for the comments that have contributed to improve the quality of this scientific communication.  

Round 2

Reviewer 2 Report

The manuscript has been improved according to the suggestions. It can be published in this current version after a check for the English language